# Experimental and Numerical Investigations of Fracture Behavior for Transversely Isotropic Slate Using Semi-Circular Bend Method

**Erqiang Li** [1,2], **Yanqing Wei** [1,*], **Zhanyang Chen** [3] and **Longfei Zhang** [2]

1   School of Civil Engineering, Luoyang Institute of Science and Technology, Luoyang 471023, China
2   State Key Laboratory for Geomechanics and Deep Underground Engineering,
    China University of Mining and Technology Beijing, Beijing 100083, China
3   School of Civil and Transportation Engineering, Henan University of Urban Construction,
    Pingdingshan 467036, China
*   Correspondence: wyq1987@lit.edu.cn

**Abstract:** Slate with inherently transverse isotropy is abundant in metamorphic rock, in buildings, and in geotechnical engineering worldwide; the tensile and shear fracture behavior of layered slate is vital to know for engineering applications. In this paper, the Brazilian and semi-circular bend (SCB) tests of layered slate were performed. The fracture characteristics of the slate were investigated by numerical simulations developed by the hybrid finite and cohesive element method (FCEM). Results showed that the measured experimental tensile strength, and mode I fracture toughness of layered slate all showed a typical V-type trend as the bedding angle increased from $0°$ to $90°$, and with divider type. The developed empirical relationship between tensile fracture toughness and tensile strength $K_{IC} = 0.094\sigma_t + 0.036$ fitted experimentally and strongly correlated. The mechanical response and fracture patterns predicted by FCEM agreed well with those of the laboratory experiments. Moreover, the shear fracture behavior and mode II fracture toughness of the layered slate were explored by systematic numerical simulations. Research results provide potential insights for further prediction and improvement of the complex fracture behavior of anisotropic rock masses for rock engineering.

**Keywords:** bedding plane; layered slate; semi-circular bend; cohesive zone model; fracture behavior

## 1. Introduction

Consideration of the influence of rock anisotropy in residential construction or underground engineering is important, and has been thoroughly demonstrated [1–5]. Transversely isotropic (TI) slate is a natural stone widely used in tunneling, building decoration, etc. TI slate with multiple bedding planes exhibits the apparent TI characteristics of thin-layered plate structure, i.e., tensile or shear failure and strong anisotropy. Such thin weak interfaces significantly affect the brittle fracture behavior and therefore cause common failure in engineering structures [1–10]. This issue is of wide concern in building construction and geotechnical engineering. Consequently, the study on fracture behavior of layered slate with developed bedding planes is of utmost importance for safe construction.

As a pronounced lithological feature, the existence of developed interfaces in sedimentary and metamorphic rocks usually causes highly anisotropic characteristics of mechanical and fracture properties. Extensive surveys have been performed to understand the failure behaviors and anisotropic responses of stratified rocks [11–15]. Cho et al. [12] investigated the anisotropic characteristics of strength through uniaxial compression and Brazilian tests for the gneiss, shale, and schist. Dan et al. [13] observed that the tensile strength and failure mode are closely related to the bedding angles by the Brazilian test for layered gneiss and other four rocks. The effect of weak interfaces on the tensile strength and failure behavior of layered sandstone, and shale were also investigated by Khanlari [14] and Mousavi et al. [15].

For TI slate, Hao et al. [7,8] conducted a series of experiments to investigate the effects of initial microcrack, bedding angle, and confining pressure, on the physicomechanical properties such as fracture angle, compressive strength, and cohesion. Gholami [16], and Alam et al. [17] focused on Brazilian and uniaxial compression tests on layered slate and obtained the failure patterns, tensile strength, and compressive strength. The effect of confining and water on mechanical properties was also investigated by Chen et al. [9]. Based on the difference of mechanical properties of layered slate, Li, Debecker, and Tan et al. [5,18–20] studied the fracture patterns by experiment and numerical simulation, concluding that the strength anisotropy has a great influence on the tensile and shear fracture.

On the other hand, numerous efforts have been engaged on the mode I (opening tensile) and mixed-mode fracture such as the edge cracked semi-cylinder disk (ECSD) and the semi-circular bend (SCB) method that are widely used in concrete, asphalt concrete, gypsum, and rocks [21–34], especially for the rocks by the SCB method because of their strengths of simplicity, minimal requirement of machining, and convenience of testing. For example, some research papers assessed the effect of bedding on the tensile fracture behaviors of layered granite [27,28], and layered sandstone [29–34]. Moreover, Wang [35], Lee [36], and Chandler et al. [37] evaluated the mode I fracture toughness of layered shales using the SCB method.

For the fracture test of TI slate, only Ulusay et al. [38] measured the mode I fracture toughness by the chevron bending method, and Alam et al. [17,39,40] carried out experimental research on fracture behavior by the wedge splitting and four/three-point bending method. However, little work has dealt with the SCB test for TI slate up to the present. The reason for this paucity is that the microstructure, mineralogy, and inherent weak planes make slates, from the aspect of specimen preparation of the experiment, very difficult to be tested for such experimental research by the SCB method. Especially, the SCB method needs complex and meticulous preparation for the tiny specimen machining by water lubrication.

Recent studies have shown that the cohesive zone model (CZM) can be employed to simulate the fracture behavior of anisotropic materials effectively [41–46]. Truong et al. [41] analyzed the failure behavior of scarf patch-repaired composite laminates under bending load by CZM. Several SCB numerical works with the CZM for characterizing the asphalt mixture's mechanical properties and fracture development of the interface crack were established successively [42–45]. Moreover, Celleri et al. [45] used the hybrid discontinuous Galerkin–CZM to simulate fracture initiation and propagation on discs in the presence of weak planes for anisotropic rock that dealt with the influence of bedding angle, and cohesion of weak interfaces. Jiang et al. [46] developed the numerical model by inserting cohesive elements into finite elements and investigated the rock fracture, penetration force, and chip shape during rock breaking with a conical pick fairly well.

Despite the importance and increased interest of mode I (opening tensile), and mode II (in-plane shear) fracturing in slate, there remains little work describing its tensile and shear fracture behavior. Compared with the amount of available data on other rocks, such as sandstone and shale, the scarcity of data on slate is appreciable, regarding its opening tensile and shear fracture toughness. This lack of data is mainly because of the difficulty of preparing laboratory SCB specimens according to the ISRM suggested methods [21,38], in which the weak bedding planes easily separate during drilling, sawing, and polishing of samples. Such samples disintegrating into semi-finished slim discs are not suitable for testing. Therefore, with the slate at hand, we observed vein SCB specimen preparations of approximately 70%, using water as a lubricant.

Additionally, the hybrid finite and cohesive element method (FCEM) is widely used to characterize the uniaxial and Brazilian test of anisotropic rocks. However, few studies have been conducted so far on crack propagation and evolution of tensile fracture in layered rocks by the SCB test under different bedding angles. In this study, Brazilian and SCB tests on anisotropic carbonaceous slate with weak bedding planes were performed systematically to investigate the tensile strength, the opening tensile fracture characteris-

tics, as well as the effect of bedding on fracture toughness of TI slate. Then FCEM was employed to simulate crack initiation and complex crack interaction when encountering bedding and fracture patterns under different bedding angles. Additionally, the specifically asymmetric semi-circular bend (ASCB) tests of shear fracture behavior for layered slate were performed by numerical simulation. The proposed empirical relationship from the experimental and numerical fitted data is significant in predicting the related fracture parameters of slate and other TI rocks.

## 2. Specimen Preparation and Experimental Method

### 2.1. Descriptions of the TI Slate

The layered carbonaceous slate samples were collected from the face of the Minxian tunnel of G75-Hai-Lan (Haikou-Lanzhou) Expressway in Gansu Province of China (Figure 1), in which their thin-bedded structure exhibited the characteristics of fragmentation and looseness and the micro-cracks and weak bedding were well developed. The average density of samples processed in the laboratory was 2.688 g/cm$^3$. X-ray diffraction analysis showed that the slate is composed predominately of quartz (43.8–56.5%), clay minerals (42.0–53.1%), and plagioclase (1.5–2.2%). The clay minerals are mainly illite (37–50%), chlorite (18–32%), illite-smectite formation (12–22%), kaolinite (8–13%), and smectite (0–3%).

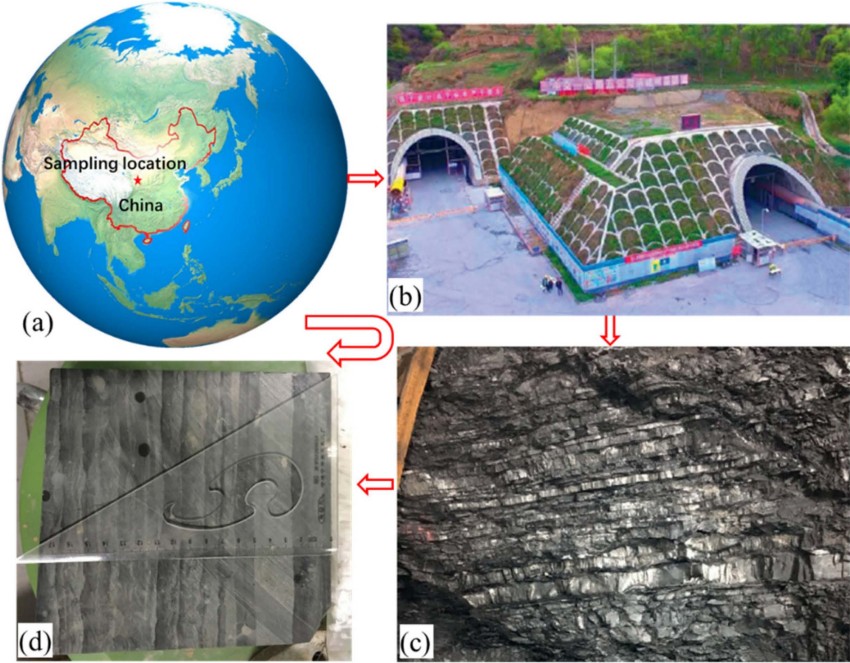

**Figure 1.** Illustration of the sampling site and the layered slate.

### 2.2. Specimen Preparation

The sample was processed into a disc specimen with a diameter of 50 mm and a thickness of approximately 25 mm under predetermined bedding angles to measure the indirect tensile strength of the TI slate by the Brazilian test. In this work, six groups of disc specimens having different bedding orientations (e.g., 0°, 30°, 45°, 60°, and 90°) to the end surface were prepared, named as BA, BB, BC, BD, BE, and BF, respectively.

The SCB test suggested by the International Society for Rock Mechanics (ISRM) was used here to measure the opening tensile fracture characteristics, and the geometrical configuration of the SCB specimen as shown in Figure 2. The radius and the thickness are R and B, respectively, the notch length and width are b and t, respectively, and the loading span is S.

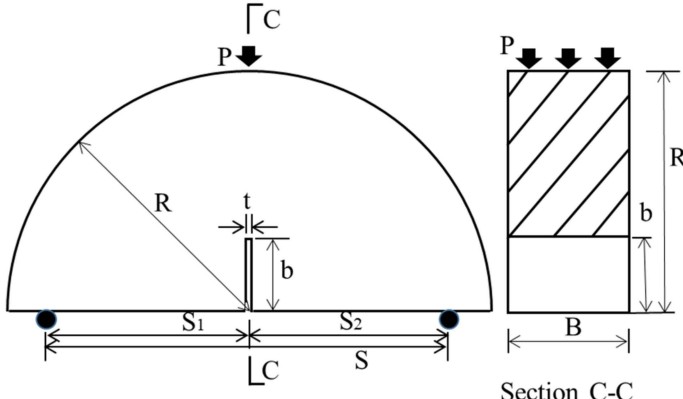

**Figure 2.** Schematic of SCB specimen geometry.

For SCB specimen preparation, a precision wire saw (0.4 mm diameter) was used to obtain semi-circular discs with a vertical straight-through notch length of about 10 mm according to the design bedding angles. The disc and SCB specimens were all transferred to natural air-drying for 60 days before testing. To study the influence of different bedding angles on the fracture characters, the angle between the bedding plane of the specimen and the horizontal plane of the Platen framework was set to be the corresponding bedding angle, as shown in Figure 3. Six groups of SCB tests were prepared, and the SCB specimens were named as MA, MB, MC, MD, ME, and MF for 0°, 30°, 45°, 60°, 90°, and a divider type, respectively.

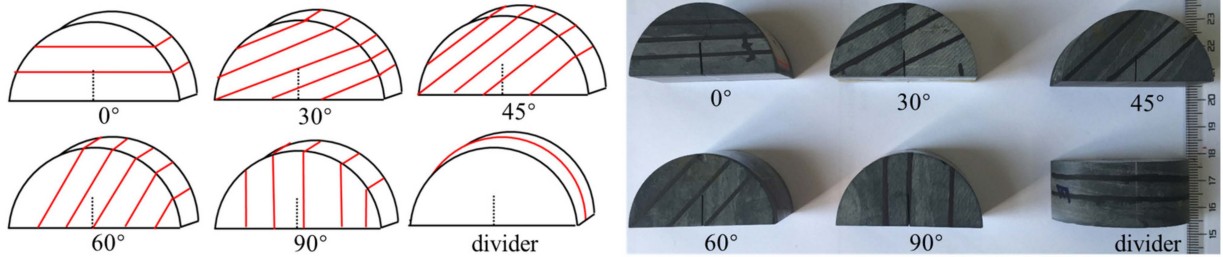

**Figure 3.** Schematic of slate SCB specimens (on the left) and the corresponding actual specimens (on the right).

### 2.3. Experimental Setup

The Brazilian tests were performed using our photoelastic precision uniaxial loading system that is equipped with a maximum axial load of 30 kN. The axial stress was applied to the displacement control with a loading rate of 0.1 mm/min, and four specimens were tested for each category in our work. During loading, the load and displacement data were automatically recorded with a load cell and two linear variable differential transformer (LVDT) sensors. The experimental setup and test process are shown in Figure 4.

The SCB tests were also conducted using the same testing system, which is mainly composed of a photoelastic loader, an extensometer displacement acquisition device, a charge-coupled discharge (CCD) high-speed camera, a graphical data collector, and supplementary light, as shown in Figure 5. The SCB loader device was constructed with a photoelastic loader consisting of one upper loading roller and two lower supporting rollers. The SCB specimens were marked with auxiliary lines in advance for symmetrical and accurate placement of the lower and upper supports, and the spacing of the supports was set to be 40 mm. The loading rate was also 0.1 mm/min, which is the relatively low loading rate that can satisfy the requirement of static crack growth suggested by ISRM [21].

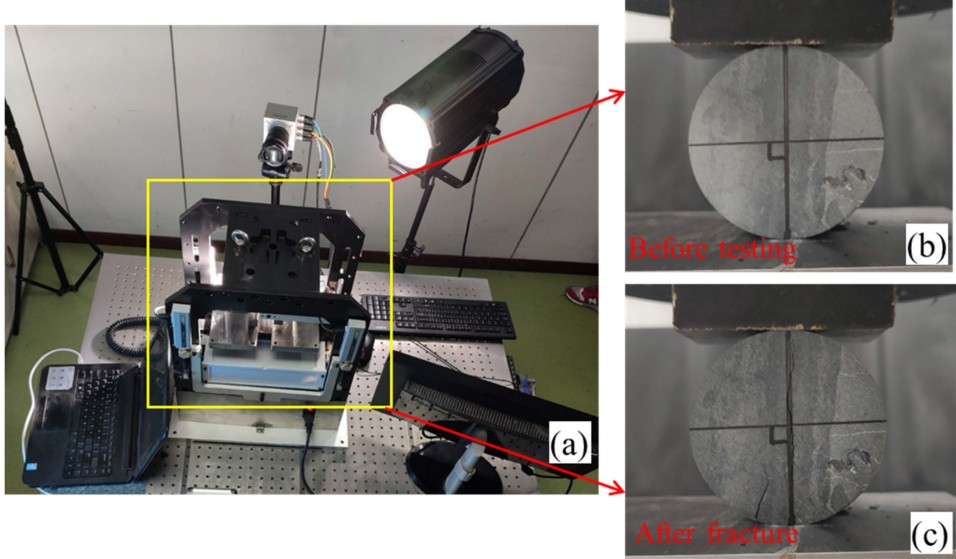

**Figure 4.** The Brazilian test system: (**a**) test setup, (**b**) sample before loading and (**c**) sample after loading.

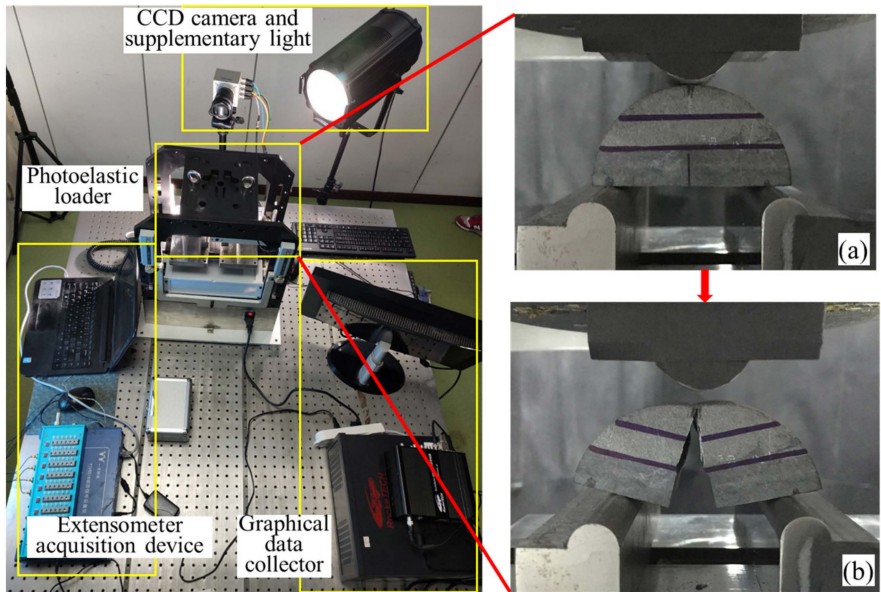

**Figure 5.** The SCB test system: (**a**) specimen loading and (**b**) specimen fracture.

## 3. Test Results

### 3.1. Tensile Strength

From the peak loads by the Brazilian tests, we can evaluate the corresponding tensile strength of the slate specimen by the analytical solution in Equation (1).

$$\sigma_t = \frac{2P}{\pi DB} \tag{1}$$

where $P$ is the peak load; $D$ is the specimen diameter; $B$ is the disc specimen thickness.

Tensile strengths of the slate specimens are shown in Table 1. When the bedding angle increased from 0° to 90°, the tensile strength decreased monotonously, and then increased sharply from 90° to the divider type, showing a typical V-type trend (Figure 6), which was in good agreement with that of other TI rocks [6,13–19]. The tensile strengths of the disc specimens with bedding angles of 0° and the divider type were greater than the oth-

ers, and the average values were 1.45 and 1.65 MPa, respectively. The measured tensile strength at the bedding angle of 90° was 0.69 MPa, which is the smallest value that can be related to rock structure and failure modes. The specimens with bedding angles of 30°, 45°, 60° were 1.09, 0.89, 0.78 MPa, respectively, the anisotropy coefficients were 2.10 (the corresponding to 0°) and 2.39 (the corresponding to divider), respectively, which indicate that the apparent anisotropic was significantly affected by the bedding.

**Table 1.** Tensile strengths of the slate specimens under different bedding angles by the Brazilian tests.

| Bedding Angle | Tensile Strength (MPa) | | | | Mean Tensile Strength (MPa) |
|---|---|---|---|---|---|
| 0° | 1.41 | 1.32 | 1.48 | 1.59 | 1.45 ± 0.114 |
| 30° | 1.12 | 0.92 | 1.10 | 1.22 | 1.09 ± 0.125 |
| 45° | 0.85 | 0.73 | 0.97 | 1.01 | 0.89 ± 0.126 |
| 60° | 0.76 | 0.93 | 0.77 | 0.67 | 0.78 ± 0.108 |
| 90° | 0.68 | 0.71 | 0.59 | 0.78 | 0.69 ± 0.079 |
| Divider | 1.64 | 1.73 | 1.49 | 1.74 | 1.65 ± 0.116 |

Note: the last column indicates the mean value of strength ± standard deviation (SD).

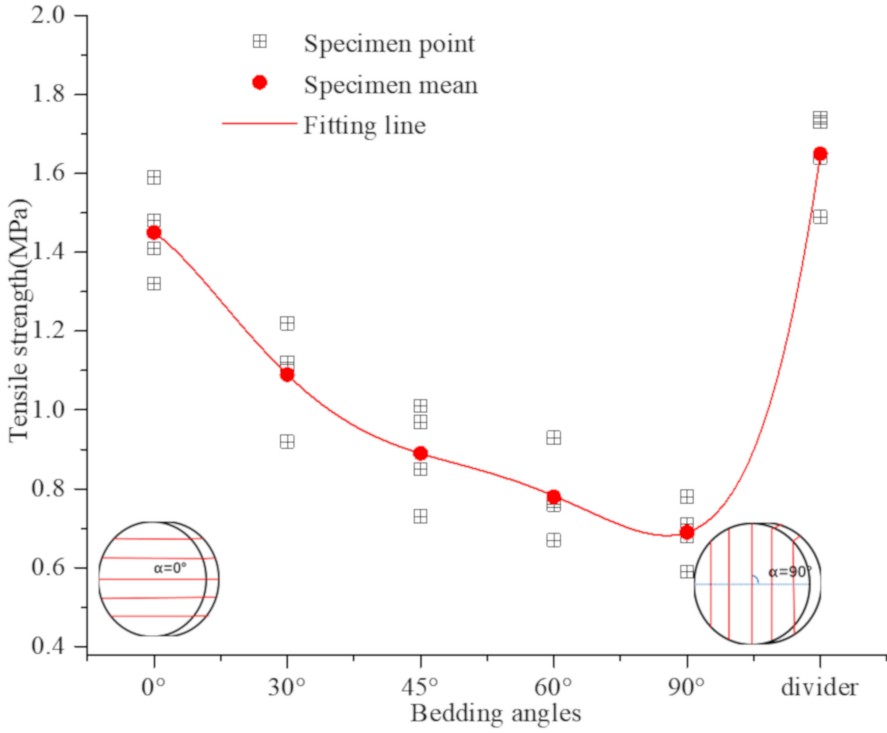

**Figure 6.** Fitting results of tensile strengths for anisotropic slate under different bedding angles.

### 3.2. Load-Displacements and Mode I Fracture Toughnesses of the SCB Tests

The load-displacement curves of SCB tests reflect constitutive characteristics, which were useful to further study the fracture toughness of the slate. In Figure 7, the load-displacements with respect to the specimens of MA-3 (0°), MB-2 (30°), MC-1 (45°), MD-1 (60°), ME-2 (90°), and MF-3 (divider) are given, which exhibit a slowly increasing section until the peak load that is followed by a dramatically falling post-failure portion. From this figure, it is readily seen that as the oriented angle to the bedding increases, the corresponding peak load progressively decreases (227.5, 192.8, 166.7, 144.3, and 136.1 N), while the peak load for the case of MF-3 (divider) is 267.9 N. Before the peak loads, almost all the load-displacement curves can be described by the linear constitutive models, excluding

the result for the divider case. During the post-failure section, the typical curves under all the bedding angles and the divider type show a rather abrupt load drop, which exhibits the characteristics of brittle failure for the slate in this work.

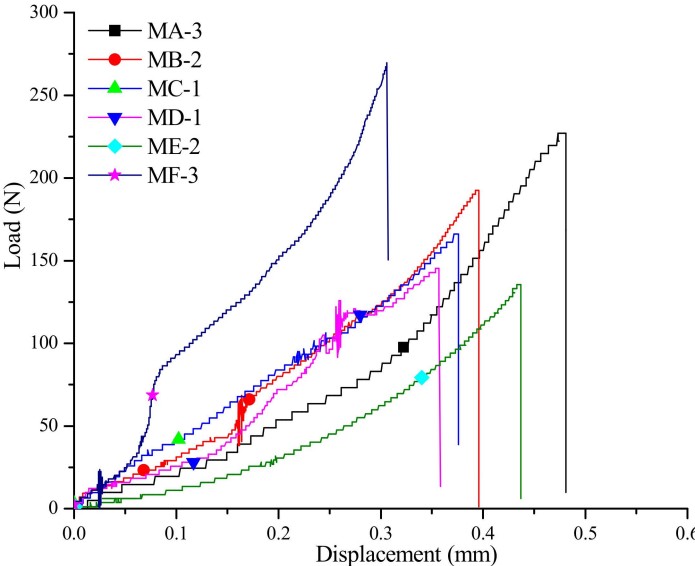

**Figure 7.** Typical load–displacement curves of the SCB specimens under different bedding angles.

According to the recommendation by ISRM [21], the mode I fracture toughness $K_{IC}$ of the slate can be evaluated by the following formulation,

$$K_{IC} = \frac{P_{max}\sqrt{\pi b}}{2RB}Y' \tag{2}$$

where $P_{max}$ is the peak load; $b$ represents the notch length; $R$ and $B$ denote the radius and thickness, respectively. $Y'$ is the dimensionless stress intensity factor under plane stress state and can be written as follows [22];

$$Y' = -1.297 + 9.516\left(\frac{S}{2R}\right) - \left(0.47 + 16.457\left(\frac{S}{2R}\right)\right)\beta + (1.071 + 34.401\,(S/2R))\beta^2 \tag{3}$$

where $\beta = b/R$, $S$ represents the span between the two bottom support rollers loaded on the SCB specimen; and $S/2R$ is the dimensionless support distance ($S/2R = 0.8$ is selected in this work).

The number of successful tests for the MA (0°), MB (30°), MC (45°), MD (60°), ME (90°), and MF (divider) is 8, 7, 7, 7, 8, and 8, respectively. By statistical analysis of the test results, $Y'$ and $K_{IC}$ can be calculated from Equations (2) and (3), and are given in Table 2.

**Table 2.** Mode I fracture toughness of the carbonaceous slate under different bedding angles by the laboratory SCB tests.

| Bedding Angle | $K_{IC}$ (MPa$\sqrt{\text{m}}$) | | | | | | | | Mean $K_{IC}$ (MPa$\sqrt{\text{m}}$) |
|---|---|---|---|---|---|---|---|---|---|
| 0° | 0.173 | 0.146 | 0.176 | 0.163 | 0.17 | 0.178 | 0.186 | 0.152 | $0.168 \pm 0.0135$ |
| 30° | 0.145 | 0.137 | 0.142 | 0.147 | 0.13 | 0.104 | 0.154 | - | $0.137 \pm 0.0112$ |
| 45° | 0.123 | 0.112 | 0.098 | 0.132 | 0.115 | 0.109 | 0.158 | - | $0.121 \pm 0.0195$ |
| 60° | 0.117 | 0.108 | 0.125 | 0.109 | 0.127 | 0.112 | 0.114 | - | $0.116 \pm 0.0070$ |
| 90° | 0.104 | 0.098 | 0.119 | 0.11 | 0.087 | 0.077 | 0.091 | 0.09 | $0.097 \pm 0.0136$ |
| Divider | 0.171 | 0.175 | 0.196 | 0.183 | 0.226 | 0.208 | 0.219 | 0.182 | $0.195 \pm 0.0207$ |

It is found in Figure 8 that the $K_{IC}$ of the divider specimens is about 0.195 MPa$\sqrt{}$m, which is the maximum among the six groups, which means that in this situation the slate specimen can have the highest fracture resistance. The slate specimens in the situation of ME (90°, i.e., short transverse direction) have the lowest fracture toughness, 0.097 MPa$\sqrt{}$m, which are with both the weakest fracture resistance and the worst tensile properties. The slate specimens in the situation of MA (0°, i.e., arrester direction), are with the toughness $K_{IC}$ of 0.168 MPa$\sqrt{}$m, which is close to that in the divider direction. This also shows that specimens with horizontal bedding have better tensile properties while the toughnesses $K_{IC}$ in the situations of MB (30°), MC (45°), and MD (60°) are 0.137, 0.121, and 0.116 MPa$\sqrt{}$m, respectively, also showing a typical V-type trend under different bedding angles which agrees fairly well with that of the layered shales measured by Wang, Lee and Chandler et al. [30–32].

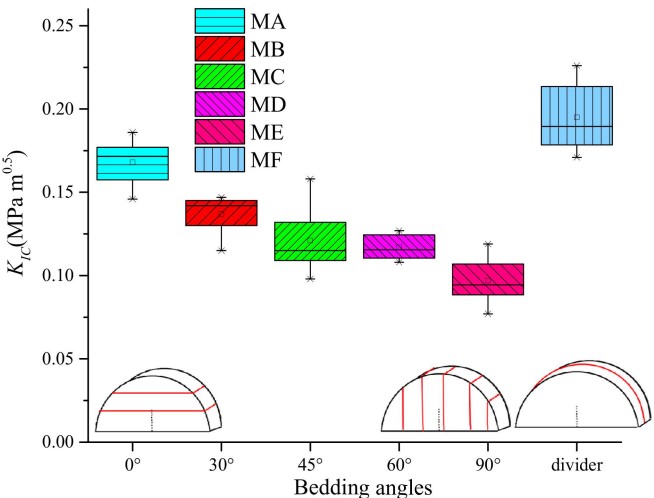

**Figure 8.** Box and whisker plot of mode I fracture toughness for different SCB specimens.

### 3.3. Empirical Relation between Fracture Toughness and Tensile Strength

To gain fracture toughness $K_{IC}$ from tensile strength $\sigma_t$ of rock, a relevant empirical relation has been found to be very useful in rock engineering. Much work has confirmed that $K_{IC}$ as a mechanical parameter of rock is associated with the tensile strength [47,48]. To date, Zhang's empirical formula $\sigma_t = 6.88K_{IC}$ derived from the statistics of numerous rock tests is widely used, but it cannot be used for slate [48] due to lack of data for slate tests. Based on the test results of carbonaceous slate specimens, we can obtain an empirical relationship between mode I fracture toughness and tensile strength, as shown in Figure 9, which is represented by an approximately linear formulation where its correlation coefficient $R^2$ is 0.99, indicating that the method of estimating rock $K_{IC}$ by $\sigma_t$ is highly fittable. The present fitting formulation can be read as

$$K_{IC} = 0.094\sigma_t + 0.036 \tag{4}$$

This fracture toughness is very crucial in evaluating the fracture behavior in rock engineering, using fracture mechanics and numerical modeling.

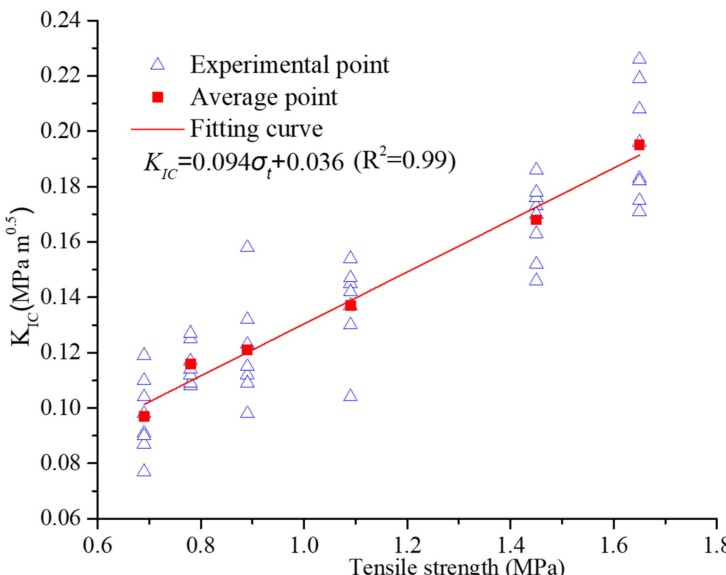

**Figure 9.** Empirical fitting relationship between tensile strength and $K_{IC}$ of the carbonaceous slate.

## 4. Numerical Modeling with Anisotropic Slate

### 4.1. CZM and Its Finite Element Formulation

The hybrid finite and cohesive element method (FCEM) by inserting cohesive elements into finite elements is widely used for applying to the fracture simulation of kinds of materials that including composites and various rocks [41–46]. When a cohesive zone formulation is employed to model cracking, it implies that the notion of a cohesive force ahead of the crack that prevents propagation is introduced. The micro-mechanisms of material deterioration and fracture are thus embedded into the constitutive law that relates the cohesive traction $\sigma$ with the local separation. Damage is restricted to evolve along the intrinsic cohesive interfaces, where cohesive elements could be randomly distributed in the mesh generated. In Figure 10, $\Delta_m^f$ and $\Delta_m$ are the critical displacement of failure and the effective opening displacement of the crack, respectively; $f_t$ represents the tensile strength (cohesive) on the crack surface.

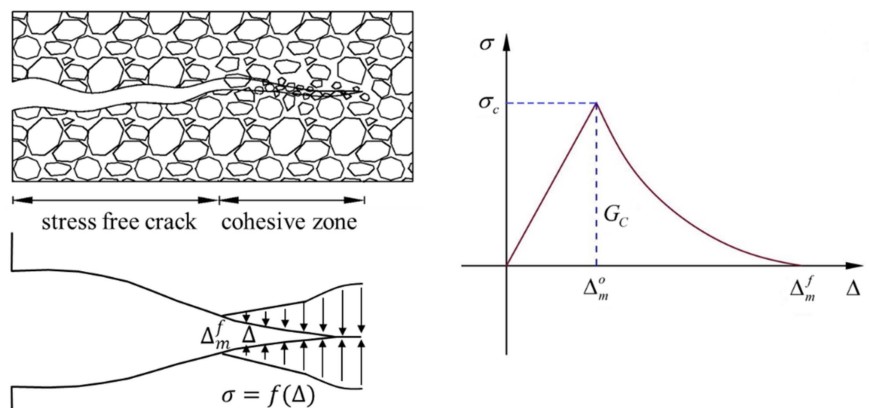

**Figure 10.** Illustration of the CZM and its form of separation law.

In the fracture model of the continuous–discontinuous method, the mode I peak strength is defined by a constant tensile strength, $f_t$, while the mode II peak strength, $f_s$, is computed according to the Mohr–Coulomb criterion. The complete breakage of the crack element and consequently the nucleation of a discrete crack are accomplished after dissipating the material fracture energy release rate, $G_c$. After breakage of the cohesive surface, the

crack element is removed from the simulation and model transition from a continuum to discontinuum is thus locally completed.

Damage initiation is referred to the beginning of deterioration of the response of a material point, which indicates the degradation of interface elements in the slate in the present work. The process of degradation begins when the stresses and/or strains satisfy certain damage initiation principles that we use. In the present work, the damage is assumed to initiate when the maximum nominal stress component ratio reaches a dimensionless value of one, which can be read as

$$\text{MAX}\left\{\frac{\langle T_n \rangle}{T_{nmax}}, \frac{T_s}{T_{smax}}, \frac{T_t}{T_{tmax}}\right\} \geq 1 \tag{5}$$

where $T_n$, $T_s$, and $T_t$ represent the normal stress and the two tangential stresses of the cohesive element, respectively; $T_{nmax}$, $T_{smax}$, and $T_{tmax}$ is the normal strength and the two shear strengths of the cohesive element. In addition, it is assumed that the cohesive element does not become damaged under pure compressions so that we introduce MacAulay parentheses < >, in which $\langle T_n \rangle$ can be read as follows;

$$\langle T_n \rangle = \begin{cases} T_n, & T_n \geq 0 \\ 0, & T_n < 0 \end{cases} \tag{6}$$

The cohesive constitutive law is completed by an appropriate equation governing the evolution of damage $D$. It is of note that $D$ remains constant if the traction on the interface is less than its current strength, if the interface is unloaded, or if $D$ reaches 1. Otherwise, $D$ has to evolve so that the strength of the interface decreases progressively from its initial value to zero as the effective displacement $\Delta_m = \left(\langle \Delta_n \rangle^2 + \Delta_s^2 + \Delta_t^2\right)^{1/2}$ increases from $\Delta_m = \Delta_m^0$ (the effective displacement at damage initiation) to $\Delta_m = \Delta_m^f$ (the effective displacement at complete failure) by an exponential form. This requires

$$D = \int_{\Delta_m^0}^{\Delta_m^f} \frac{T_{eff}}{G_c - G_0} d\Delta \tag{7}$$

$$\Delta_m^f = 2G_c / T_{eff}^0 \tag{8}$$

$$T_{eff} = \left(\langle T_n \rangle^2 + T_s^2 + T_t^2\right)^{1/2} \tag{9}$$

Consequently, we have the following formulations: The relationship between stress components and damage variable is as follows;

$$T_n = \begin{cases} (1-D)E_n\Delta_n, & \Delta_n \geq 0 \\ E_n\Delta_n, & \Delta_n < 0 \end{cases} \tag{10}$$

$$T_s = (1-D)E_s\Delta_s \tag{11}$$

$$T_t = (1-D)E_t\Delta_t \tag{12}$$

where $E_n$, $E_s$, and $E_t$ are the normal and the two tangential stiffnesses, respectively; $\Delta_n$, $\Delta_s$, and $\Delta_t$ the normal displacement and the two tangential displacements, respectively.

The Benzeggagh–Kenane (BK) law is particularly useful when the critical fracture energies during deformation, purely along the first and second shear directions, are the same, i.e., $G_{SC} = G_{TC} = G_{IIC}$, which is read as follows;

$$G_C = G_{IC} + (G_{IIC} - G_{IC})(G_S/G_T)^\gamma \tag{13}$$

where $G_I$, $G_{II}$, and $G_{III}$ stand for the fracture energies from mode I to mode III, respectively; $G_s = G_{II} + G_{III}$, and $G_T = G_I + G_S$, and $\gamma$ is a material constant.

### 4.2. Numerical of the Brazilian Tests

### 4.2.1. Modelling Procedure

The CZM mentioned previously was implemented in the FCEM framework of Abaqus/Explicit software, the 3D computational model for the disc specimen was established first, the model characterizes weak plane spacing from top to bottom mainly of 9.1–12.5 mm by partitioning the model into different blocks under the design bedding angle. Second, the elastic solid element (C3D4) was employed in the model to simulate the mechanical behavior of the slate, and the bedding planes were simplified, and were described by the inserted cohesive elements of an in-house computer program using MATLAB code to insert cohesive elements (COH3D6) into the initial finite element model, and a typical 3D computational model with the bedding angle of 0° as indicated in Figure 11.

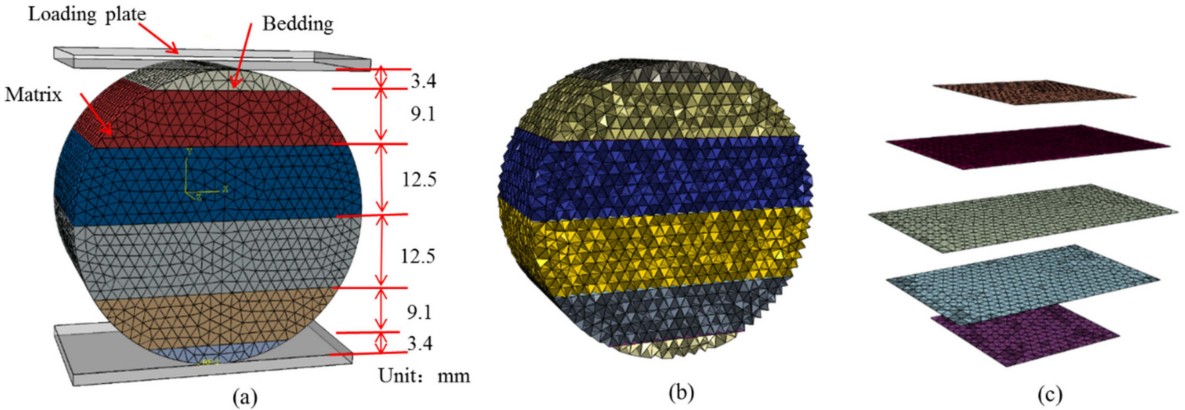

**Figure 11.** 3D computational model of carbonaceous slate under bedding angle of 0°, (**a**) model setup, (**b**) embedded cohesive elements for the matrix, and (**c**) embedded cohesive elements for the bedding planes.

It is emphasized that the same cohesive elements inserted in the model were used to simulate failure or fracture; different mechanical properties were considered to address either the behavior of the bedding plane or the intrinsic performance of the slate matrix. The elastic parameters required for the matrix are Young's modulus $E$ and Poisson's ratio $\nu$, which are given by $E$ = 5.2 GPa and $\nu$ = 0.2. Since there was not a perfect approach to directly obtain the model parameters of the cohesive elements from the conventional experiments of TI slate, we used the trial and error method to fit the experimental data and the macroscopic response of the specimens to obtain the parameters in the present work. Using the above-mentioned process, the constitutive parameters of the cohesive elements were inserted in the weak plane and the internal matrix of the slate. All of the parameters used in the CZM simulations are listed in Table 3.

**Table 3.** The input parameters of the slate specimens in CZM simulations.

| Cohesive Elements | Matrix | Bedding Planes |
|:---:|:---:|:---:|
| $E_n$ (N/mm³) | 1150 | 950 |
| $E_s$ (N/mm³) | 1350 | 1150 |
| $T_{nmax}$ (MPa) | 2.0 | 1.0 |
| $T_{smax}$ (MPa) | 4.0 | 1.5 |
| $G_{IC}$ (N/mm) | 0.015 | 0.008 |
| $G_{IIC}$ (N/mm) | 0.15 | 0.08 |

### 4.2.2. Numerical Model Verification

To verify the correctness of the numerical model, Brazilian tests were performed by FCEM under bedding angles of 0°, 30°, 45°, 60°, 90°, and divider type. The experimental and numerical load–displacement curves are shown in Figure 12, where their shapes are fairly consistent and the corresponding ultimate failure patterns are also fairly well identified for the corresponding bedding cases. Additionally, for the typical bedding angles of 0°, 90°, and divider, the tensile strengths of specimens were 1.43 MPa (vs. the lab test mean value of 1.45 MPa), 0.69 MPa (vs the lab test mean value of 0.69 MPa), 1.61 MPa (vs the lab test mean value of 1.64 MPa), respectively. In such results, the peak load of slate is reduced with the increase of a bedding angle of 0°–90°, as shown in Figure 12a–e, while increasing sharply with a bedding angle of 90° to the divider type, as in Figure 12e,f, showing a V-type trend similar to the results of laboratory Brazilian test in this work, This indicates that the bedding direction significantly affects the tensile strength due to the relatively anisotropic mechanical properties of the weak structural planes among the layered slate.

For the macro failure patterns of all cases in the present work, as loading is continuously applied, there are many micro-cracks that emerged near the two loading ends, which contribute to small-scale crush. For the bedding angles of 0° and 90°, the fracture paths run through the two loading ends, and the shear failure occurs between the horizontal bedding planes when the bedding angle is 0°, as shown in Figure 12a. When the bedding angle is 90°, straight tensile failure is formed between the vertical bedding planes near the loading end, as shown in Figure 12e. Figure 12b–d, it is of note that for bedding angles of 30–60°, the numerical results often partially present shear fracture along the inclined bedding planes. The simulated failure patterns are fairly consistent with the Brazilian test results of slate in the literature [5,6,18–20]. For the divider model, the vertical principal crack appears along the vertical loading line and accompanies a small arc, which is consistent with the failure patterns of typical experimental specimens, as shown in Figure 12f.

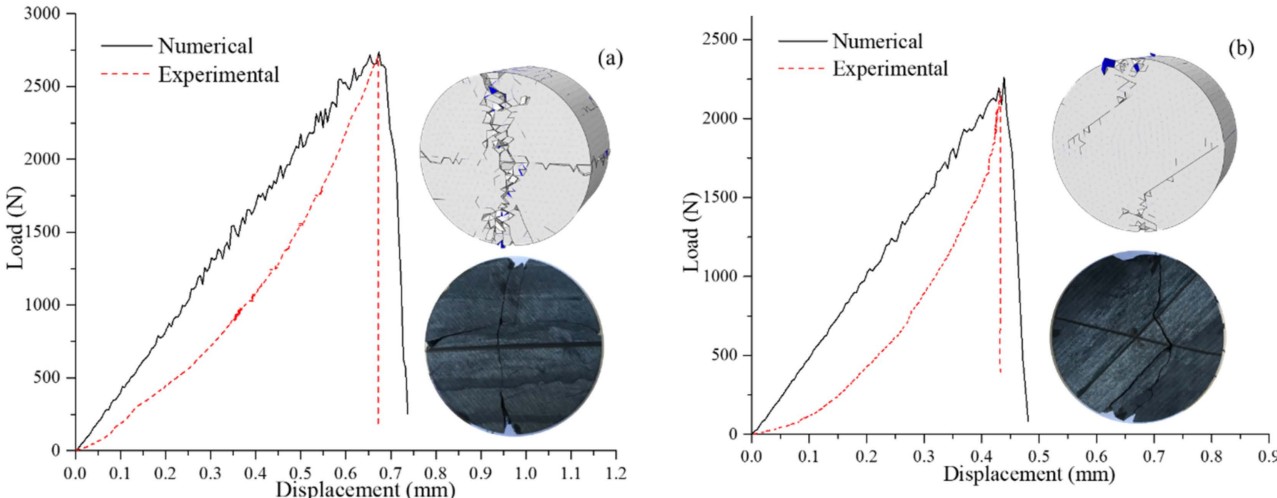

**Figure 12.** *Cont.*

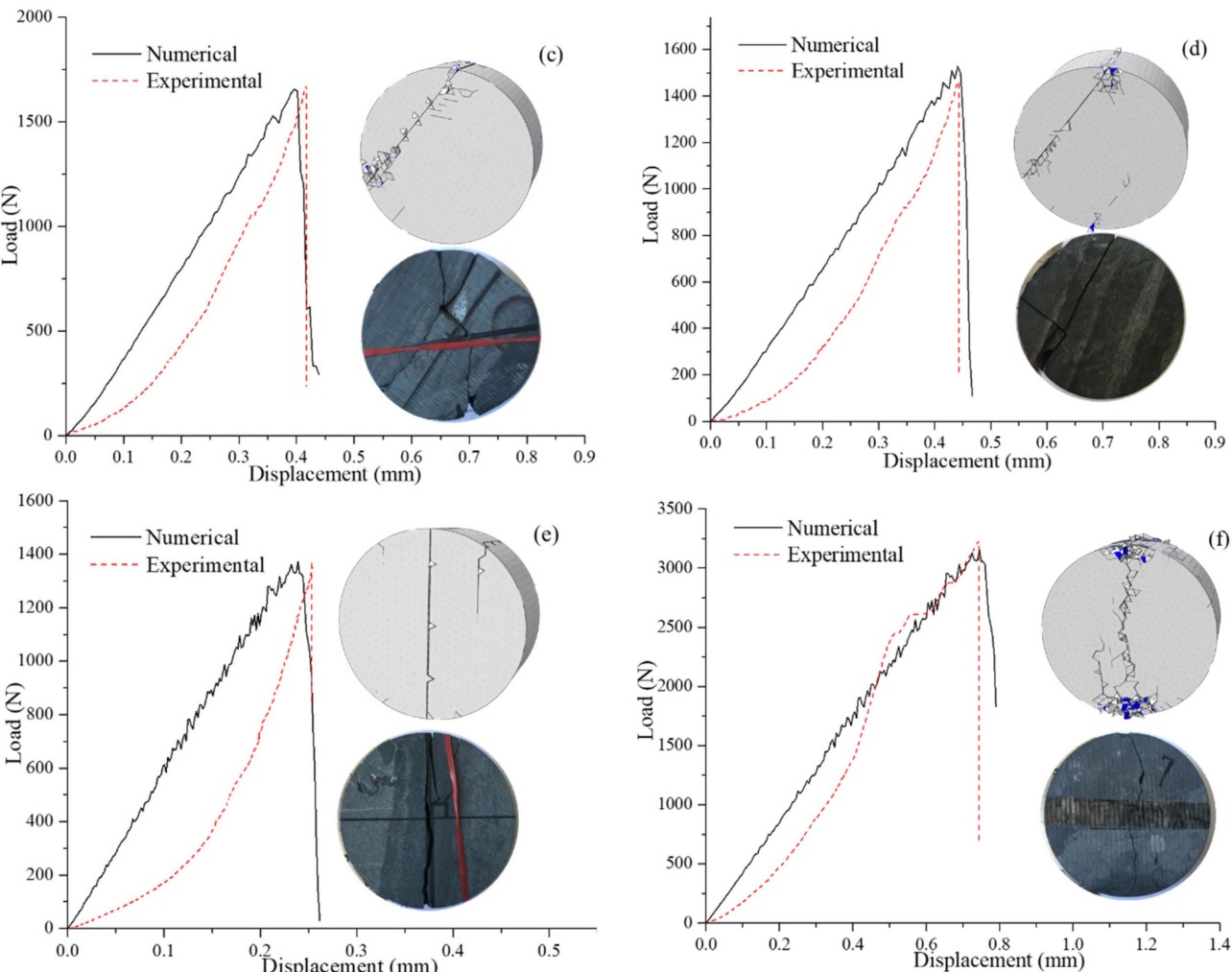

**Figure 12.** Comparison of the load-displacement curves and final failure patterns for the experimental and numerical Brazilian tests with the bedding angles of (**a**) 0°, (**b**) 30°, (**c**) 45°, (**d**) 60°, (**e**) 90°, and (**f**) divider.

### 4.3. Results and Discussion of the SCB Tests

#### 4.3.1. SCB Numerical Model Setup

According to the typical bedding distribution of the carbonaceous slate specimen, the 3D computational models for SCB simulation were established; the weak planes (veins) with the zero-thickness cohesive elements are bedded into specimens according to the predetermined bedding angles and spacing between 4.0–9.0 mm, in which the spacing between interfaces is mainly 6.0–7.0 mm in the computational model. The dimensions of the semi-circular model are 50 mm in diameter and 25 mm in thickness with a straight-through notch of 10 mm length and 0.4 width from the middle bottom of the numerical model. Six numerical models having different bedding orientations (e.g., 0°, 30°, 45°, 60°, 90°, and divider) were established systematically, and the typical specimen for the bedding angle of 0° is indicated in Figure 13. Three bedding planes are within the specimen model by partitioning the model into different blocks while their spacings from top to bottom are 4.0, 7.0, 7.0, and 7.0 mm, respectively. The number of solid elements (C3D4) and cohesive elements (COH3D6) in the model are 52736 and 102696, respectively. It is of note that bedding arrangements need to be subdivided leading to different model blocks, which have a slight impact on the mesh generation under different bedding angles and divider type.

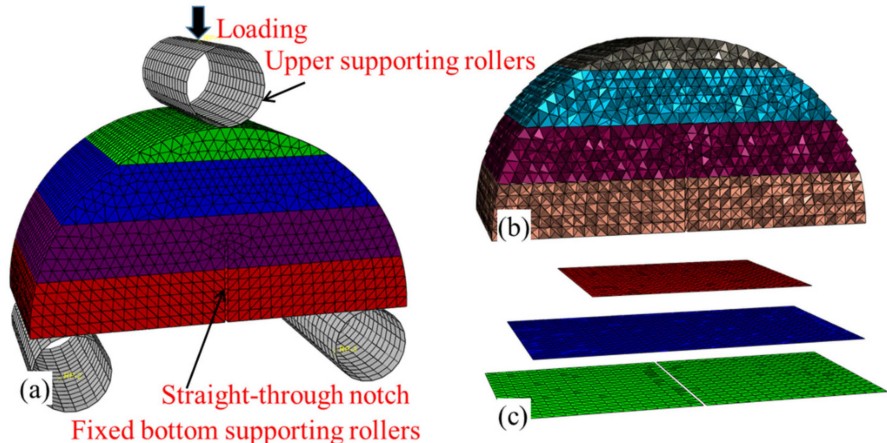

**Figure 13.** SCB computational SCB model of the carbonaceous slate under bedding angle of 0°, (**a**) model setup, (**b**) embedded cohesive elements for the matrix, and (**c**) embedded cohesive elements for the bedding planes.

In the numerical SCB tests, one upper rigid loading roller and two lower (bottom) rigid supporting rollers were employed to load the specimen using the displacement control that moves vertically downwards at a constant rate of $3.0 \times 10^{-6}$ mm/step. This ensures that the numerical specimens are under quasi-static states during numerical testing, whereas the two bottom supporting rollers at both ends are fixed and their span is 40 mm. Again, the appropriate input parameters of the slate specimens in CZM simulations were applied here, as given in Table 3.

### 4.3.2. SCB Results and Discussion

This section presents the typical experimental and numerical results of anisotropic slates with different bedding angles, which cover both numerical and experimental load-displacement and typical failure patterns of the slate SCB specimens at bedding angles of 0° and 60° as detailed in Figures 14 and 15. The cases of bedding angles of 30°, 45°, 90°, and divider are concisely presented in Figure 16. It is of note that the zone of interest (ZOI) is set up to present in part the detailed fracture cracks.

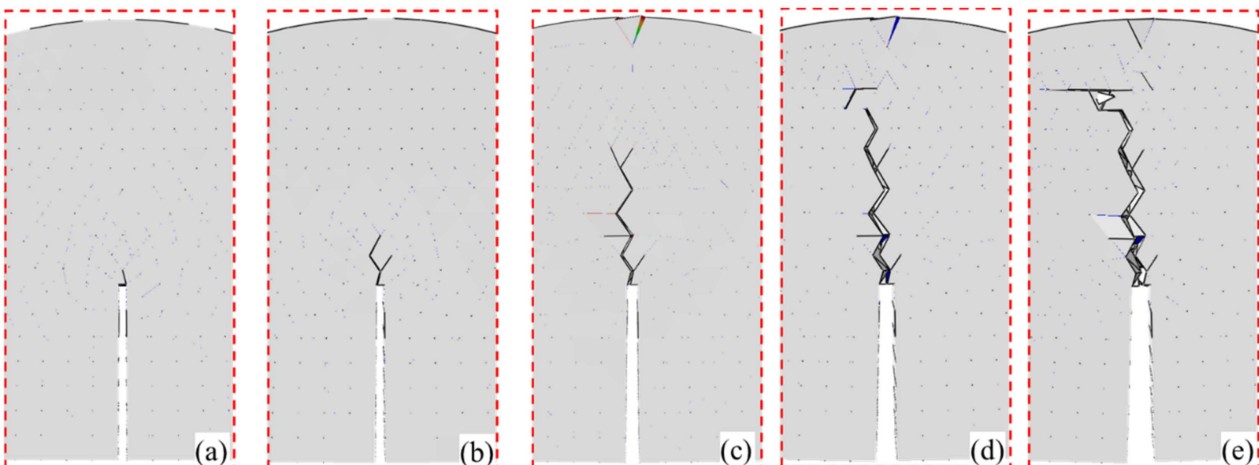

**Figure 14.** *Cont.*

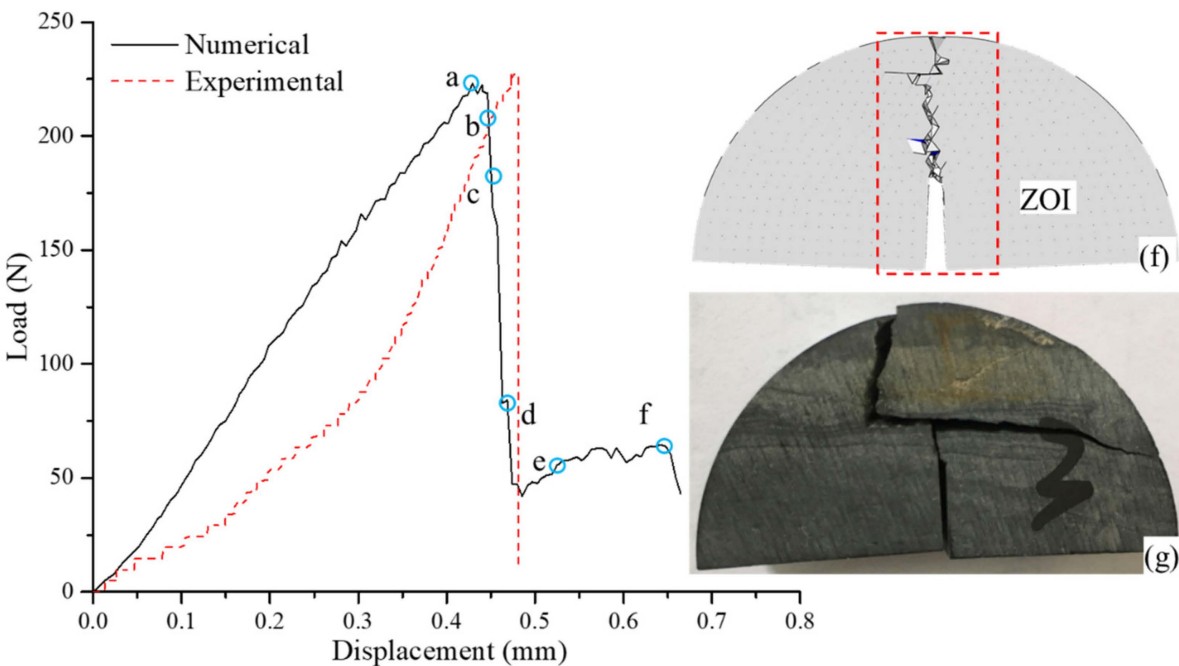

**Figure 14.** Comparison of the tensile fracture propagation and load–displacement measured from numerically and experimentally with an inclined angle of 0°, (**a–f**) corresponding to the points marked in the load–displacement curve, and (**g**) fracture failure of the representative testing specimen.

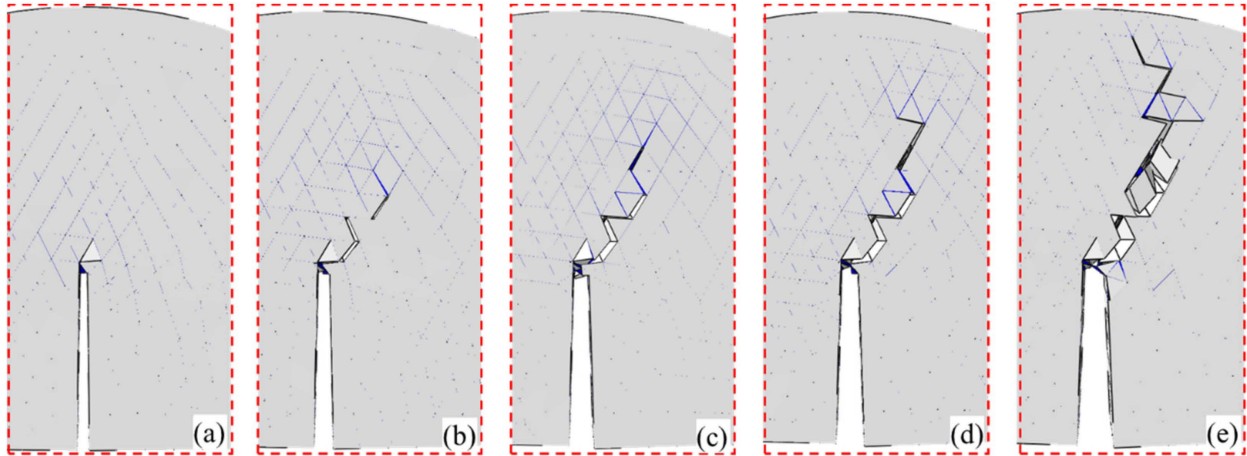

**Figure 15.** *Cont.*

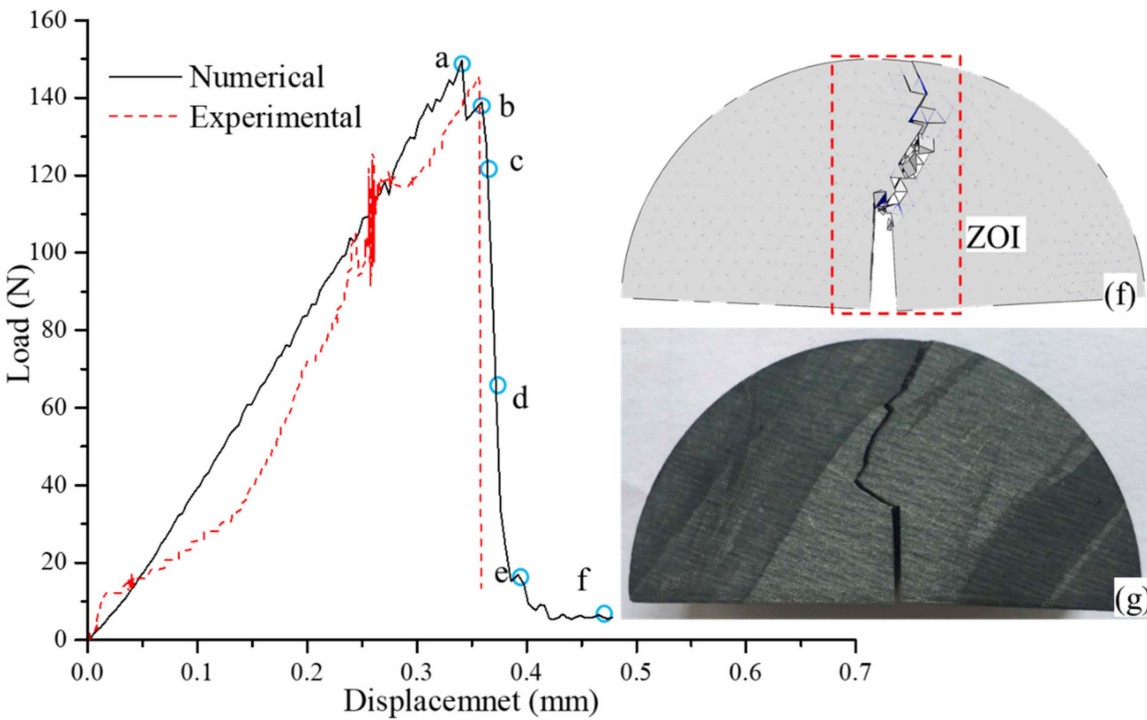

**Figure 15.** Comparison of tensile fracture propagation and load–displacement measured numerically and experimentally with an inclined angle of 60°, (**a**–**f**) corresponding to the marked points (**a**–**f**) of load–displacement curve, and (**g**) the fracture pattern of the representative slate specimen.

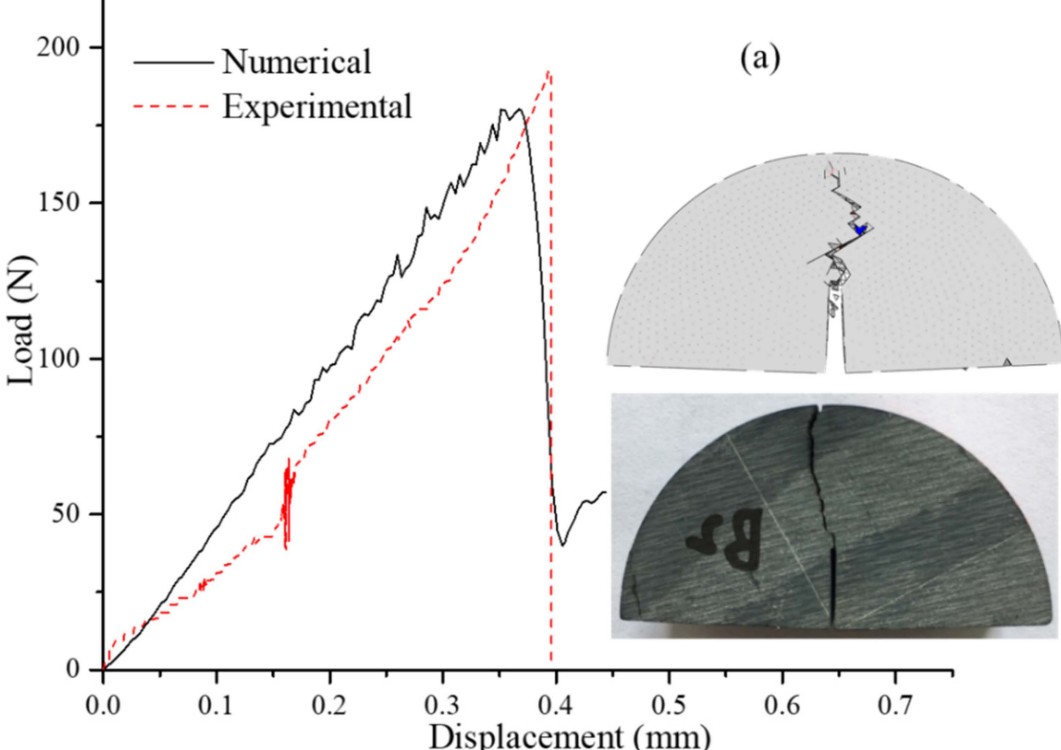

**Figure 16.** *Cont.*

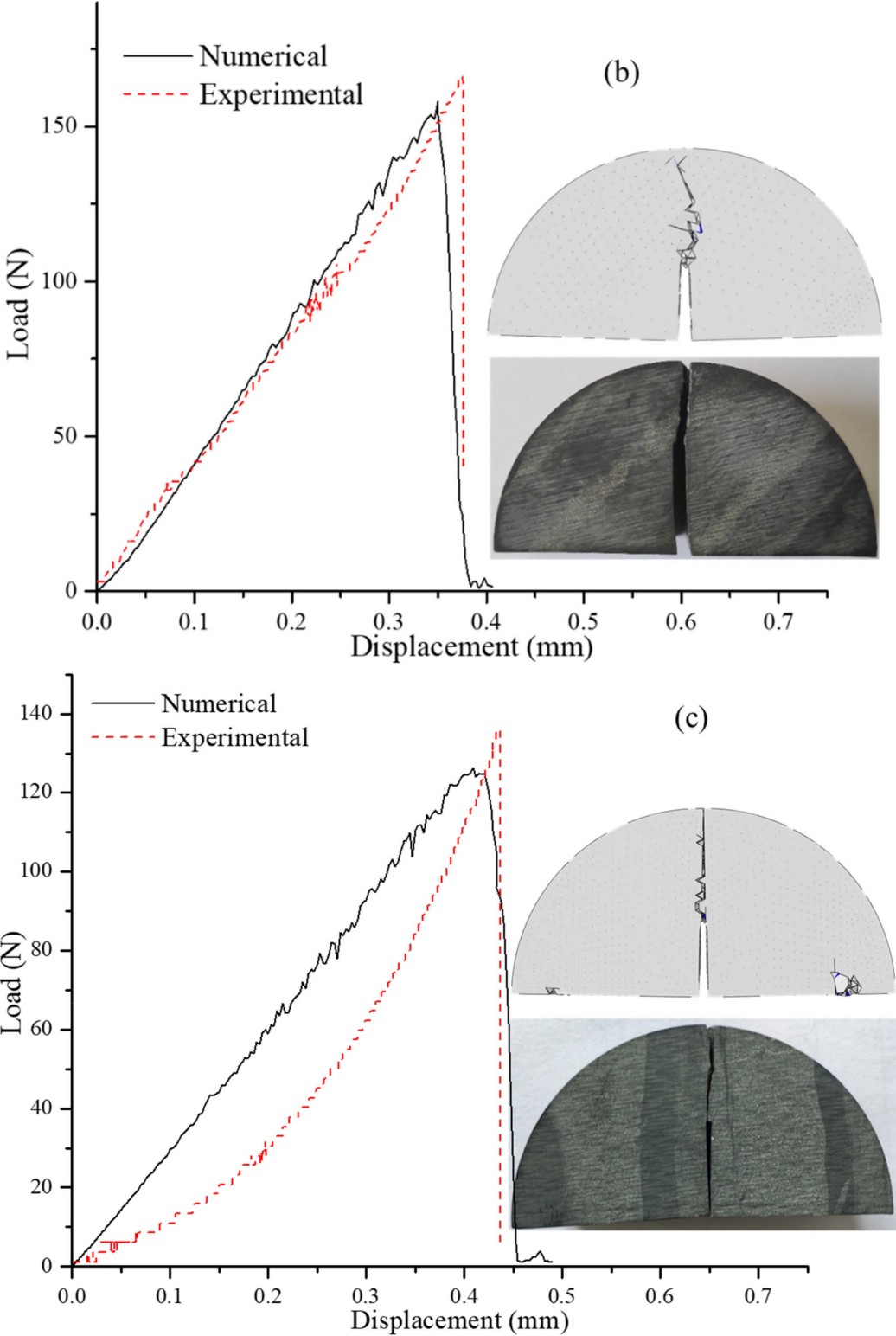

**Figure 16.** *Cont.*

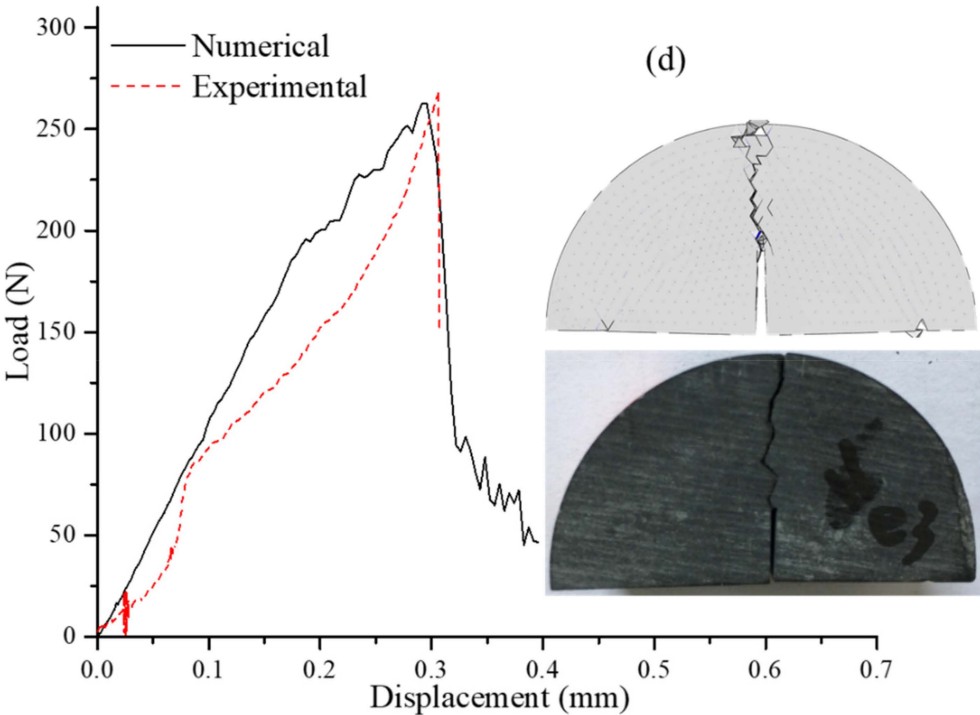

**Figure 16.** Comparison of the fracture patterns and load-displacement curves measured the numerically and experimentally with bedding angles of (**a**) 30°, (**b**) 45 °, (**c**) 90°, and (**d**) divider.

For the case of the inclined angle of 0°, the tensile crack first initiates from the notch tip at the small oscillation at the peak load (Figure 14a), then the tensile crack extends rapidly as the load–displacement curve shows a rather abrupt load drop (Figure 14b) and the fracture propagation is unstable. A few cracks are generated inside the middle bedding plane as the principal opening fracture approaches the upper bedding plane and then it propagates in a zigzag manner towards the upper loading point (Figure 14c,d). Furthermore, due to the concentration of compressive stress induced by the three-point bending load on the SCB specimen, the principal fracture develops further towards the loading end (Figure 14e,f). It can be seen that the fracture evolution is almost dominated by tensile fracture, and the mechanical response and failure patterns exhibit some differences between the numerical simulations and the experimental tests. The reason is that the numerical model cannot completely take into account all of the heterogeneity of the real specimen with the micro-structure because the latter is a natural material whose micro-defects, micro weak interfaces, and micro-damage cannot be perfectly known in advance. In general, the specimens are with characteristics of significant brittle fracture that exhibit a sharp drop in loading, and evolution of interlayer cracks after the crack initiation finishes abruptly. This also matches the instantaneous 'bang' fracture during the laboratory tests.

For the case of the inclined angle of 60°, there also exists a tensile crack that initiates from the notch tip at the peak load (Figure 15a), but the fracture propagates towards the middle bedding and then rapidly travels along the bedding direction for a small distance before a snake-like macro fracture travels up to the top (see Figure 15b–f). Overall, the mechanical response and failure patterns of the numerical results agree fairly well with the typical experimental results, the fracture behavior of the numerical and typical experimental specimens were all very brittle as the load dropped sharply, while the interaction of the opening tensile fracture between the bedding and matrix occurred instantaneously after the peak load.

Figure 16 illustrates the fracture patterns and load–displacement curves measured from numerically and experimentally with the bedding angles of 30°, 45°, 90°, and divider

type. The overall responses of the curves all show brittle failure as a sharp drop, and each numerical result agrees fairly well with the typical experimental results, respectively.

Figure 16a,b clearly shows that the dominant fracture initiates from the notch tip and ultimately propagates to the upper loading point. While intersecting with the bedding planes, the crack usually diverted into the bedding planes and kinked back into the host matrix after a certain propagation distance along the weak planes. It was observed that the deflected tensile fracture along the bedding plane was very short under an inclined angle of 45°, but a long deflected tensile fracture along the bedding plane occurred at about 30°. The results show that the fracture patterns agree fairly well with the testing specimens.

Figure 16c clearly illustrates the failure pattern of the main fracture crack under an inclined angle of 90°, which is mainly manifested as a vertical straight-through crack from the notch tip to the upper loading end, which is essentially the same as the postmortem for the tested specimen. In Figure 16d, the numerical result of the divider shows that the crack propagation presents a small zigzag from the notch tip upward to the upper loading end, which is completely consistent with the fracture path of the typical tested specimen.

From Figure 14 to Figure 16, it can be readily seen that as the oriented angle to bedding increases, the corresponding numerical peak load progressively decreases (223.5, 180.1, 158.4, 149.7, and 127.8 N), while the peak load for the case of divider type is 262.5 N, and the calculated numerical $K_{IC}$ is shown in the following Figure 17.

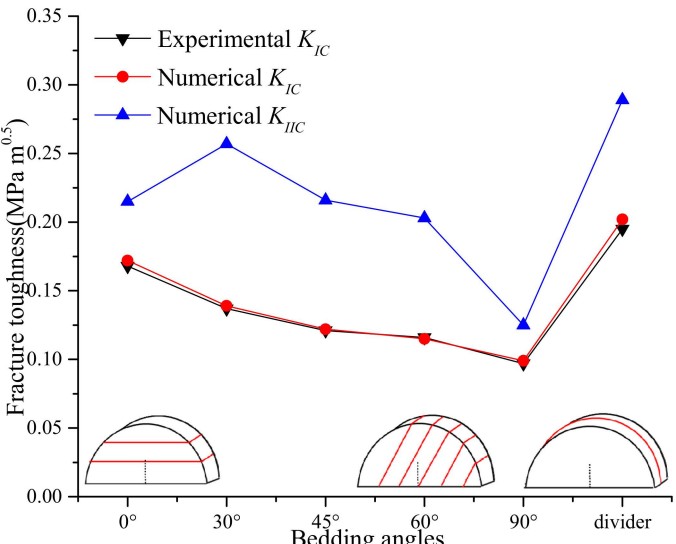

**Figure 17.** The variation of $K_{IC}$ and $K_{IIC}$ under different bedding angles obtained by experimental and numerical tests.

## 5. Shear Fracture Behavior by FCEM

Prior work has documented fracture behavior of slate such as large strength anisotropy in both laboratory experiments and FCEM, while the tensile fracture patterns and failure processes of slate have been carefully compared experimentally and numerically. However, these studies did not pay attention to the shear fracture that is also fundamental in evaluating the failure properties of slate. Thus, the objective of this section is to examine the mode II fracture toughness ($K_{IIC}$) and its shear fracture behavior by conducting a series of asymmetric semi-circular bend (ASCB) numerical tests for slate with appropriate values for $S_1$ and $S_2$ (see Figure 2) under the same loading rate as above [25,26,49]. According to reference [49], the $K_{IIC}$ of the TI slate can be evaluated by the following formula;

$$K_{IIC} = \frac{P_{max}\sqrt{\pi b}}{2RB} Y''$$ (14)

where $Y''$ is the dimensionless stress intensity factor under the plane stress state that correlates with a/R, $S_1$/R, and $S_2$/R; here its value is 1.7822.

We found that for the oriented angle of the bedding of 0°, 30°, 45°, 60°, 90°, and divider type, the corresponding numerical peak loads for the mode II loading were 852.4, 1018.9, 855.2, 804.4, 496.3, and 1142.4 N, respectively. Figure 17 depicts the variation of fracture toughness $K_{IC}$ and $K_{IIC}$ with different bedding angles obtained from the experimental and numerical tests. As indicated in the figure, we found that the values of experimental $K_{IC}$ are consistent with the numerical $K_{IC}$. However, the shear fracture toughness changes in a complicated way. As the bedding angle increase from 0° to 90°, the numerical $K_{IIC}$ values of the layered slate change sequentially as follows: 0.215, 0.257, 0.216, 0.203, and 0.125 MPa$\sqrt{m}$, then sharply increase to 0.289 MPa$\sqrt{m}$ for the divider type. Accordingly, the numerical $K_{IIC}$ exhibits an increasing trend first from 0° to 30° then decreases from 30° to 90°, while ultimately sharply increasing from 90° to divider type. Therefore, it can be concluded that the present numerical values of fracture toughness $K_{IC}$ and $K_{IIC}$ predicted by the FCEM are fairly good.

The complete shear fracture patterns and internal fracture trajectories of simulated ASCB specimens are shown in Figure 18. In the cases for bedding angle of 0° under mode II loading, the crack initiates from the notch tip with an angle relative to the vertical notch line, and the fracture propagates in a stable manner before rapidly swaying when the crack encounters the horizontal bedding, and then it travels up to the top boundary of the loading (see Figure 18a). With the increase of inclined bedding angle from 30° to 60° covered divider type, the fracture patterns of the simulations are also essentially the same, while the fracture patterns are very complex with greater interaction between weak planes and the fracture, which not only enhance the peak load but also affect the interaction pattern of the fracture plane (see Figure 18b,d,f). However, with a bedding angle of 90° under mode II loading, the fracture pattern is illustrated as a vertical straight-through crack from the notch tip to the upper loading end (see Figure 18e), which may be mainly affected by the artificial weak planes in the computational ASCB model.

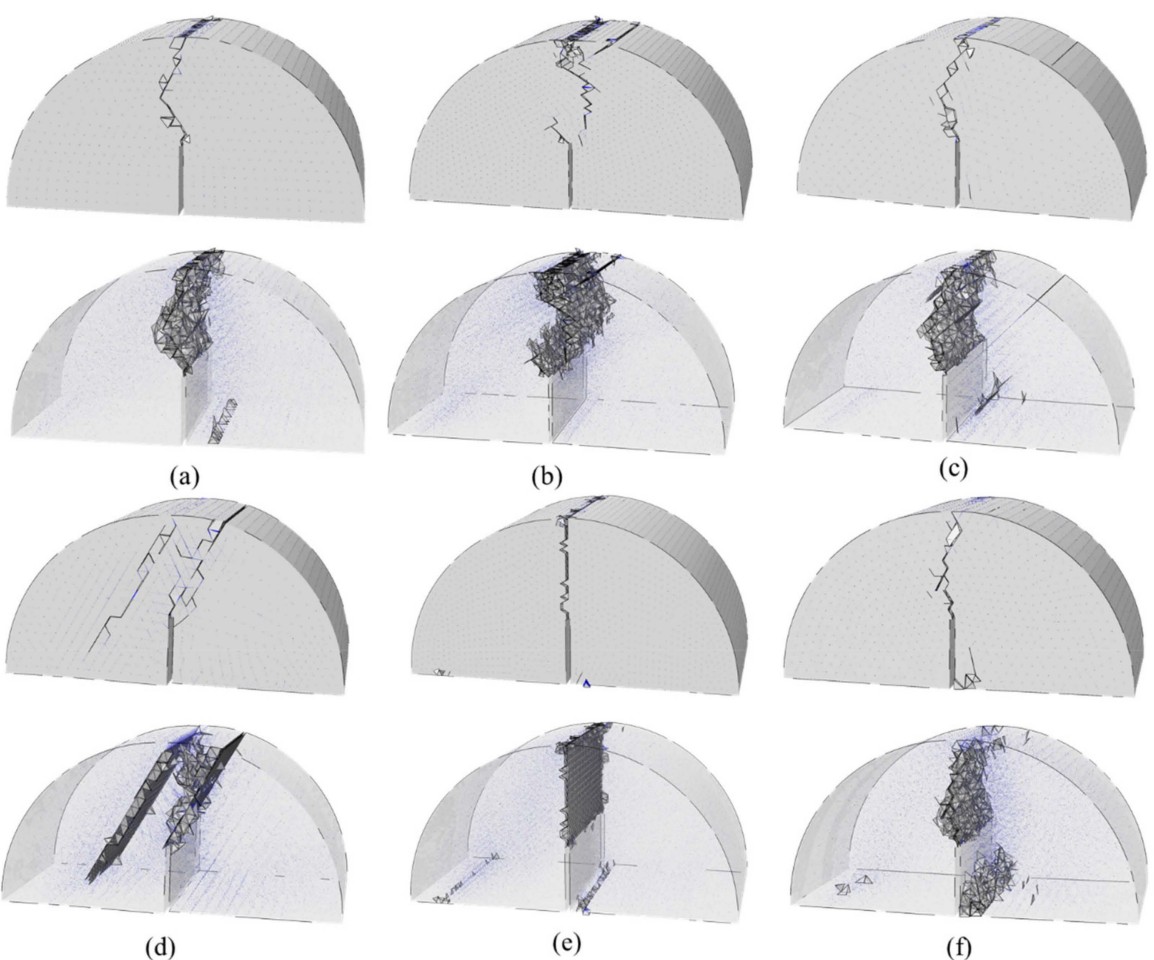

**Figure 18.** The final 3D failure patterns of the numerical ASCB tests by FCEM with the bedding angles of (**a**) 0°, (**b**) 30°, (**c**) 45°, (**d**) 60°, (**e**) 90°, and (**f**) divider.

## 6. Conclusions

A series of the Brazilian and SCB tests of layered carbonaceous slate under multiple bedding angles were conducted to study the tensile strength, mode I fracture characteristics, and fracture processes of TI slate. Corresponding numerical simulations provided insights into the opening tensile interaction between the bedding and matrix in the layered slate. The mechanical response and fracture patterns by FCEM with different bedding angles were compared with the corresponding experimental results. The mode II fracture toughness and its shear behavior were explored by numerical simulation. Based on the experimental and numerical results, the following conclusions were drawn.

The tensile fracture properties of anisotropic slate with different bedding angles were investigated by the Brazilian and SCB tests, which reflect the anisotropic effect of bedding on the fracture characteristics of layered slate.

A developed empirical relationship of $K_{IC} = 0.094\sigma_t + 0.036$ between mode I fracture toughness and tensile strength was proposed, which provides a convenient way to predict the mode I fracture toughness of layered rock.

The mechanical response and fracture patterns predicted by FCEM agree fairly well with the experimental results.

Through systematic in-depth numerical simulations for shear loading by FCEM, the mode II fracture toughness and its failure behavior were explored numerically.

Overall, this work provides potential insights into the tensile and shear fracture characteristics of anisotropic rock masses from both experiments and FCEM modeling. This

combined method can be used to explain and predict the complex mixed fracture behavior of rock and associated similar problems in rock engineering.

**Author Contributions:** Conceptualization and writing—original draft preparation, E.L.; methodology and writing—review and editing, Y.W.; software and formal analysis, Z.C.; investigation and data curation, E.L. and L.Z. All authors have read and agreed to the published version of the manuscript.

**Funding:** This research was funded by the National Key R&D Program of China (No. 2016YFC06009 01), the Key research Project of higher education institutions in Henan Province (Grant NO. 23B5600 11) and the Key Science and Technology Research Projects of Henan Province (Grant NO. 2221023203 76). Besides, it was funded by Research Fund for high level talents of Luoyang Institute of Science and Technology (NO. 21010714 and NO. 21010625) and Luoyang Institute of Science and Technology education and teaching reform research project (Grant NO. 2021JYZK-020).

**Institutional Review Board Statement:** Not applicable.

**Informed Consent Statement:** Not applicable.

**Data Availability Statement:** Not applicable.

**Acknowledgments:** First of all, thanks are due to Jili Feng, Dejian Li, and Yingjun Li for their valuable help on testing suggestions and professional discussions. Moreover, we would like to extend our thanks to Lu Chen, Mingyuan Zhang, Zhenqun Qi, Chunxiao Li, Jiashu Wang, and Hao Qi for their support of experimental works. Last but not least, the authors would also like to thank Zhenyu Zhang and Yanwei Chen for their assistance in the technical writing of this paper. The authors specially thank Xingyu Zhang in his support for refining our paper.

**Conflicts of Interest:** The authors declare no conflict of interest.

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
