# Peer review of "Experimental and Numerical Investigations of Fracture Behavior for Transversely Isotropic Slate Using Semi-Circular Bend Method"

_applsci, doi:10.3390/app13042418_

Round 1

Reviewer 1 Report

Brazilian bending test and semicircle bending test have been carried out on layered slate. The fracture characteristics of layered slate were studied by numerical simulation using FCEM method. The author's work is innovative to some extent, but there are some problems in the logic of the article. Therefore, My recommendation flag for this manuscript is set to "major revision".

 (1) How to prepare the sample in the laboratory test, such as how to make the laminated surface, is not very clear, could you please give more details

 (2) In Section 2.3, why was the Brazilian splitting test performed, what was the significance of doing the Brazilian splitting test, and what were the data derived from the test?

 (3) Figure 6 unit error, please modify

 (4) In the numerical simulation in Section 4.2, since it is a composite rock sample composed of layers with different mechanical properties, the structural plane between layers is very important. It is suggested to add the description of rock layer connection and strength parameters in this section. For example, the bedding planes strength selection basis, etc.

 (5) In Chapter 4, there is no explanation on why the work in Section 4.1, 4.2 and 4.3 is done, which leads to some problems in logic and defects in structure.

 (6) In the numerical simulation in sections 4.2 and 4.3, only the test results were described without the author's own analysis, so it is suggested to increase the analysis and summary of the experimental results. This was also true in the laboratory experiments described above.

 (7) The whole paper mainly focuses on bedding angle, but there are few explanations and analyses about bedding angle in the analysis.

Reviewer 2 Report

The paper is well written and completed. The tensile and shear behaviors are considered. However, some minor revision may be required: 

1. Please check the structure and English of the paper.

2- In the empirical formulation KI is related to tensile strength what about the numerical fracture toughness.

3- How does the size of the crack affects KI in this case?.

4- I think it is better to compare the results with the other numerical methods such as XFEM and PFC too. You may check the following papers.

Numerical simulation of the effect of bedding layer on the tensile failure mechanism of rock using PFC2D, Structural Engineering and Mechanics, An Int'l Journal, 2019, 69 (1), 43-50.

L Zhou,V Sarfarazi,H Haeri,P Ebneabbasi, M Fatehi Marji &M Hassannezhad Vayan, 2021, A new approach for measurement of the fracture toughness using the edge cracked semi-cylinder disk (ECSD) concrete specimens, Mechanics Based Design of Structures and Machines, An International Journal,

5- the conclusion part of the paper may be improved. most of the conclusions are quite evident and trivial. however, the main conclusions may be given in bullet form and  separated from the first and last paragraphs.  

Round 2

Reviewer 1 Report

Has been modified, suggest English progress polish after receiving!